extaille; extinction; extirpation; Poisson process; rediscovery; time series

**Corresponding author:**
Rachel McCrea;
Email: r.mccrea@lancaster.ac.uk

# Inferring species extinction from sighting data

Rachel S. McCrea[1] [iD], Thomas Cheale[2], Eduard Campillo-Funollet[1] and David L. Roberts[3]

[1]School of Mathematical Sciences, Fylde College, Lancaster University, Lancaster, UK; [2]School of Mathematics, Statistics and Actuarial Science, University of Kent, Canterbury, UK and [3]Durrell Institute of Conservation and Ecology, School of Anthropology and Conservation, University of Kent, Canterbury, Kent, UK

## Abstract

1. Understanding whether a species still persists, or the timing of its extinction is challenging, however, such knowledge is fundamental for effective species management.
2. For the vast majority of species our understanding of their existence is based solely on sighting data that can range from museum specimens and clear photographs, through vocalisations, to markings and oral accounts.
3. Here we review the methods that have been developed to infer the extinction of species from a sighting record, providing an understanding of their assumptions and applications. We have also produced an RShiny package which can be used to implement some of the methods presented in the article.
4. While there are a number of potential areas that could be further developed, the methods reviewed provide a useful tool for inferring species extinction.

## Impact statement

When a species has gone unseen for a period of time, the question naturally arises as to whether the species still persists. Knowing whether a species is extant or extinct is fundamental for effective conservation. A series of methods have been developed for the inference of extinction based on sighting data (e.g. museum specimens, photographs, markings and oral accounts). Here we provide a review of the different methods describing the underlying assumptions and important considerations when applying these methods. To increase the accessibility of these methods we provide an RShiny package with instructions for its application. While these methods have most frequently been applied to extinctions of species, these end-point estimations have wider relevance having been applied in the context of archaeology, geological stratigraphy, phenological studies, and phylogenetics, as well as more recently the estimation of origination in archaeology, epidemiology, and geology.

## Introduction

We are now entering a time of immense environmental upheaval with a multitude of factors (e.g. habitat degradation, climate change, over-exploitation) driving species to extinction as a result of human activities. This has led some to suggest we are entering the 6th mass extinction event (Ceballos et al. 2015), with extinction rates orders of magnitude greater than background extinction rates. Knowing if a species is extinct is therefore not only fundamental to the conservation of species through effective resource allocation but also to our understanding of how the Earth is changing. The problem, however, is that extinctions are very rarely observed and therefore must be inferred given the available data.

According to the IUCN Red List criteria (IUCN 2012), the gold standard by which threat assessments are undertaken for species, a species may be listed as Extinct (EX), if there is no reasonable doubt that the last individual has died. In essence, the population size has reached zero. However, there is often uncertainty as to whether a species is extinct due to the lack of direct observations, and so more recently, the category Critically Endangered (Possibly Extinct) (CR (PE)) was developed (IUCN Standards and Petitions Committee 2022).

Uncertainty as to whether a species is extinct largely arises from the lack of direct observation of the extinction which itself is the result of a lack of data. For the vast majority of species, their persistence is only known through sightings, whether direct (e.g. specimens or photographs) or indirect (e.g. footprints, nests, scrapes, oral accounts). This is also the case for formerly more common species that have undergone a decline, as when a species approaches extinction, data often becomes scarce.

Prior to the early work of Solow (1993a,b), little consideration had been given to the development of statistical methods for inferring historical extinctions (i.e. extinctions since

1500); although McArdle (1990) laid the foundation for using probabilistic arguments for extinction declarations. It is, however, important to note that there is considerable literature in palaeo-biology research on extinction inference, although this specifically refers to the inference of the timing of extinction rather than the question of whether the species is extinct.

Early works (see Statistical methods for examples of this work) are based on the question, given the distribution of sightings $(t_1,..., t_n)$ through time, what is the probability another sighting would not have occurred between the last sighting $(t_n)$ and a specified date (e.g. today's date)? From the initial work of Solow (1993a), more complex models have been developed that incorporate potentially important factors, such as assumptions around the shape of the decline (e.g. Solow 1993b), sighting uncertainty (e.g. Solow et al. 2012) and underlying sampling effort (e.g. McCarthy 1998).

The state-of-the-art of inferring species extinction has been reviewed in a number of previous papers, e.g. Solow (2005) and Boakes et al. (2015). The methodological developments in this area have typically followed the application of well-established Bayesian or classical statistics principles to an explicit statistical model of the sighting record. We present a review of the literature around the development of methods for inferring historical extinctions (i.e. extinctions since 1500), notable extensions and future challenges

## Statistical methods

It is important to first understand two terms frequently used within the extinction modelling literature, specifically 'sighting' or 'sighting event', and 'sighting record'. In this literature, a 'sighting' or 'sighting event' refers to a single observational data point related to the existence of the species. These, as mentioned, can be anything from a museum specimen or clear diagnostic photograph, to markings and oral accounts. A 'sighting record', however, is a series of sightings of a particular species, rather than a single event; that is not to say that some species are only known from a single specimen and therefore such a sighting would also constitute the sighting record (Roberts and Jarić 2020).

Let $t_1 < t_2 < \ldots < t_n$ be the times at which a species is sighted during an observation period $(0, T)$. Suppose that the studied species become extinct at time $T_E$, then we might be interested in determining whether extinction has occurred, i.e. whether $T_E < T$. Two statistical paradigms exist for inference about extinction:

*Classical/Frequentist methodology:* Classically, an assessment could be made of the hypothesis test $H_0$: $T_E \geq T$ versus $H_1$: $T_E < T$. The null hypothesis in this case corresponds to the species being extant and the alternative hypothesis corresponds to the species being extinct. It is possible to calculate a probability ($p$-)value which corresponds to the probability of observing the sighting record under the null hypothesis. If the $p$-value is small this means that there is evidence to reject the null hypothesis and infer that the species is indeed extinct. The issue of deciding upon an appropriate threshold (significance level, $\alpha$) to draw this conclusion is of course challenging, and there is considerable current literature on the subjectivity of p-values – see for example Wasserstein and Lazar (2016). Alternatively, the focus might be on the estimation of $T_E$ and equivalently it is possible to construct the upper bound of a $100(1 - \alpha)$% confidence interval for $T_E$ of the form $(t_n, T_E^U)$. Note that this formulation is a one-sided confidence interval with the last sighting being the lower bound. It is also possible to construct a two-sided confidence interval, as in Roberts and Solow (2003), where the lower bound could be greater than the time of the last sighting.

*Bayesian methodology:* In a Bayesian framework, it is possible to evaluate the posterior distribution of the species being extant as:

$$P(T_E > T | t_1, \ldots, t_n) =$$

$$\frac{P(t_1, \ldots, t_n | T_E > T) P(T_E > T)}{P(t_1, \ldots, t_n | T_E > T) p(T_E > T) + P(t_1, \ldots, t_n | T_E \leq T)(1 - P(T_E > T))}$$

where $P(t_1, \ldots, t_n | T_E > T)$ denotes the likelihood of observing the sighting data given the species is extant and $p(t_1, \ldots, t_n | T_E \leq T)$ denotes the likelihood of observing the sighting data given the species is extinct by $T$. $p(T_E > T)$ denotes the prior probability that the population is extant.

The Bayes factor is defined to be the ratio of the likelihood of the sighting data under $T_E > T$ and $T_E \leq T$, respectively:

$$B(t_1, \ldots, t_n) = \frac{P(t_1, \ldots, t_n | T_E > T)}{P(t_1, \ldots, t_n | T_E \leq T)}$$

This represents the ratio of the posterior to prior odds in favour of species being extant. A large value of $B(t_1,..,t_n)$ represents substantial evidence that the species is extant whilst small values indicate evidence for extinction.

## Inferring extinction

Regardless of which inferential approach is taken, the methods for inferring extinction which we present here rely on the general assumption that the sightings, $t_1,..., t_n$ can be modelled as realisations of a point process which is defined by a rate function denoted by $\lambda(t)$. The point process typically used is the non-stationary Poisson process, and the rate $\lambda(t)$ is interpreted to be the instantaneous rate of sightings at time $t$, as described in Solow (2005). Within this context, $\lambda(t)$ can be considered to be proportional to the abundance of the species and the sampling effort to sight the species, however, we will later describe methods that have been developed to separate these two processes.

Using this model structure requires a number of stringent assumptions to be made. The sightings must be made in continuous time and must be independent of one another. We discuss the potential violation of these assumptions and other aspects of the implementation of these methods in Important model considerations section.

Using the properties of the Poisson process, the number of sightings in a fixed time period $(0,T)$ has a Poisson distribution with mean $\Lambda(T) = \int_0^T \lambda(u) du$. The number of sightings in the period $(0, T)$, denoted $n$, does not contain any information about extinction time so it is reasonable to condition on it. Conditional on $n$, the sightings are independent and identically distributed over the time period with probability density function $f(t) = \frac{\lambda(t)}{\Lambda(T)}$. Thus models, defined by $f(t)$ can be proposed and under the null hypothesis of an extinction time, $T_E$ this is equivalent to the determination of the endpoint of a distribution.

## Parametric models

Unsurprisingly, early proposals for modelling sighting data made a number of simplifying assumptions. In particular, the stationary Poisson process, which assumes sightings are uniformly distributed over time, such that $f(t) = 1/T$ was addressed in both a classical and Bayesian framework in Solow (1993a). A truncated exponential formulation, which allows the sighting rate to decline over time was

**Table 1.** $p$ denotes the $p$-value, such that the null hypothesis $H_0{:}T_E \geq T$ can be rejected at the significance level $\alpha$ if $p < \alpha$

| Approach | p | $\hat{T}_E$ | $\hat{T}_E^U$ |
|---|---|---|---|
| Uniform[1] | $\left(\frac{t_n}{T}\right)^n$ | $\frac{n+1}{n}t_n$ | $\frac{t_n}{\alpha^{1/n}}$ |
| Truncated exponential[2] | $\frac{F(t_n)}{F(T)}$ | $t_n + \frac{\sum_{i=0}^{[s/t_n]}(-1)^i\binom{n}{i}(s-it_n)^{n-1}}{n(n-1)\sum_{i=0}^{[s/t_n]-1}(-1)^i\binom{n-1}{i}(s-(i+1)t_n)^{n-2}}$ | † |
| End point estimation[3] | $\frac{t_n-t_{n-1}}{T-t_{n-1}}$ | $t_n + (t_n - t_{n-1})$ | $t_n + \frac{1-\alpha}{\alpha}(t_n - t_{n-1})$ |
| Adapted end point estimation[4] | $\frac{\frac{t_{n-1}}{n-1}+\gamma}{\frac{t_{n-1}}{n-1}+\gamma+(T-t_n)}$ | – | $t_n + \left(\frac{t_{n-1}}{n-1}+\gamma\right) \times \frac{1-\alpha}{\alpha}$ |
| Weibull extreme value[5] | $exp\left(-k\left[\frac{T-t_n}{T-t_{n-k+1}}\right]^{1/\hat{v}}\right)$ | $\sum_{i=1}^{k}w_i t_{n-i+1}$ | $\frac{t_n-c(\alpha)t_{n-k+1}}{1-c(\alpha)}$ |

$F(x) = 1 - \sum_{i=1}^{[s/x]}(-1)^{i-1}\binom{n}{i}\left(1-\frac{ix}{s}\right)^{n-1}$ where $s = \sum_{i=1}^{n}t_i$ and denotes the integer part. † denotes that the upper confidence limit has to be calculated numerically. $\gamma$ denotes a coefficient of trend in sighting intervals (the average change in length of intervals between each two consecutive sightings) and is given by $\gamma = \frac{\sum_{i=2}^{n-1}([t_{i+1}-t_i]-[t_i-t_{i-1}])}{n-2}$. $\hat{v} = \frac{1}{k-1}\sum_{i=1}^{k-2}\frac{t_n-t_{n-k+1}}{t_n-t_{i+1}}$, $w = (e'Y^{-1}e)^{-1}Y^{-1}e$ where e is a vector of k 1's and Y is the symmetric $k \times k$ matrix with element $Y_{ij} = \frac{\Gamma(2\hat{v}+i)\Gamma(\hat{v}+j)}{\Gamma(\hat{v}+i)\Gamma(j)}$ for $j \leq i$, and $c(\alpha) = \left(\frac{-\log\alpha}{k}\right)^{\hat{v}}$.

[1]Solow (1993a).
[2]Solow (1993b).
[3]Solow and Roberts (2003)
[4]Jarić and Ebenhard (2010).
[5]Presented in Solow and Roberts (2003) with mathematical detail given in Solow (2005).

presented in Solow (1993b) and further developed in Solow and Helser (2000). The truncated exponential model is appropriate in the case when the sighting rate is proportional to population size which, prior to extinction, declines exponentially. Rout et al. (2009) provided a Bayesian framework for the determination of whether extinction has occurred under this declining sighting rate model, but within the context of eradication of invasive species.

A substantial number of modifications have been made to these basic models with either constant or declining sighting rate. For example, McCarthy (1998) derived a model which can account for changes in collection effort and Roberts and Jarić (2020) showed how this could be adapted for data from a single specimen. McInerny et al. (2006) modified the approach by incorporating sighting rate into the model. Explicitly, their method generates a probability that another sighting will occur given the previous sighting rate and the time since last observation. These sighting rates can then be used to contrast species, even when sighting data may have been collected at different times.

### Non-parametric models

The approaches described so far are parametric, i.e. a statistical model, defined in terms of a small number of parameters, has been adopted via the specification of the probability density function, $f(t)$. Alternatively, it is possible to use a statistical method in which the data are not presumed to come from prescribed models. This approach is referred to as a non-parametric approach. Solow and Roberts (2003) propose a non-parametric test for extinction, based on the general endpoint estimation method of Robson and Whitlock (1964). This approach performs well when the sighting rate is constant, however will have lower power than the corresponding parametric approach. Jarić and Ebenhard (2010) modified this non-parametric test to account for trends in the sighting interval. Unlike the parametric approaches we have presented, the non-parametric approach does not require knowledge of the start time of the sighting interval or number of sighting observations, and inference is performed with just knowledge of the properties of the later sightings. This property, as explained in Solow (2005), motivates the use of the property that the joint distribution of the $k$ most

recent sightings is the Weibull extreme value distribution (Coles 2001). Solow and Roberts (2003) developed an approach, based on optimal linear estimation (referred to as the OLE approach in the extinction literature) to estimate $T_E$ and derived an upper confidence bound and p-value for inferring extinction. The method was used to estimate the extinction time of the Dodo, *Raphus cucullatus.*

Table 1 collates the p-value, upper $100(1 - \alpha)\%$ confidence bound and where available, the estimator of extinction time, $\hat{T}_E$, for the constant sighting rate (Uniform) model, declining sighting rate (Truncated exponential) model, non-parametric approach based on end point estimation and the corresponding adapted version which accommodates trends in sighting rate and the extreme value distribution approach. To make the methods of Table 1 more accessible we have collated them within an RShiny app, available at https://tommy-cheale-shinyapps.shinyapps.io/New_ext_pap/.

Bayesian inference requires an appropriate choice of prior probability of extinction (Solow 2016a, b) and also requires the choice of a prior distribution for the rate of the Poisson process, $\lambda(t)$. One possibility is to use a non-informative prior, or alternatively expert opinion can be used to construct an informative prior. Whichever approach is taken, it is prudent to assess prior sensitivity to avoid inadvertent influence on the conclusions from the analysis. Depending on the choice of prior probabilities it can be possible to derive an explicit form for the Bayes factor.

Under an assumption of constant sighting rate and non-informative prior on the rate of the Poisson process and a prior which assumes all values of $T_E$ are equally likely, the Bayes factor, derived in Solow (1993a) is given by:

$$B(t_1,\ldots,t_n) = \frac{n-1}{\left(\frac{T}{t_n}\right)^{n-1} - 1}.$$

These assumptions were relaxed in Rout et al. (2009) to accommodate a decreasing sighting rate, such that $\lambda(t) = mt^{-a}$, where $m$ is a constant and $a \in (0, 1)$. When $a = 0$ this simplifies to the constant sighting rate model. The corresponding Bayes factor now generalises to:

$$B(t_1,\ldots,t_n) = \frac{n(1-a)-1}{\left(\frac{T}{t_n}\right)^{n(1-a)-1} - 1}$$

### Uncertain and certain sightings

The Bayesian framework also provides a flexible mechanism by which to incorporate uncertainty in sighting records. This principle was first introduced in Solow et al. (2012) and further explored in Kodikara et al. (2018). Following the notation proposed in Kodikara et al. (2018), the first (termed Model 1) assumes that the sighting record is divided into two parts with the division based on the unknown extinction time, $T_E$. Valid sightings in the period before extinction follow a stationary Poisson process with rate $\Lambda$ and invalid sightings follow a stationary Poisson process with rate $\Theta$. Over the second time period, all sightings are invalid and follow a stationary Poisson process with rate $\Theta$. Let $j$ denote the number of valid uncertain sightings before $T_E$. If $n_c$ and $n_u$ denote the number of certain and uncertain sightings respectively, there are then $n_c + j$ valid sightings in the interval $(0, T_E)$ and $n - (n_c + j)$ in valid sightings in $(0, T)$. It is then possible to construct the likelihoods for the valid and invalid sightings in these time periods and to infer extinction from the observed sighting data.

Model 2 assumes that certain sightings follow an independent Poisson process with rate $M$, while valid uncertain sightings follow a stationary Poisson process with rate $\Lambda$ and invalid sightings follow a stationary Poisson process with rate $\Theta$. The Bayes factors corresponding to these models do not have closed-form solutions and so need to be calculated numerically. These two modelling approaches were found to differ in conclusion for an application to the ivory-billed woodpecker and particular sighting sensitivities were explored in Kodikara et al. (2018). Their finding was that each model was sensitive to certain sightings within the data set and it is not possible to ascertain which one to use for any given application.

The modelling approaches for accounting for certain and uncertain sighting data had typically made the assumption of sighting data arising from constant rate Poisson processes. Kodikara et al. (2021) relaxed this assumption by extending the models to accommodate non-homogeneous Poisson processes. The use of the non-homogeneous Poisson process offers considerable flexibility, for example, this model can accommodate species sighting records where the population size and/or sighting generation process changes through time. Interestingly, they also framed the extinction time as a change-point for uncertain sightings, since uncertain sightings before the extinction time will consist of both valid and invalid sightings, while after the extinction time, only uncertain sightings will be observed.

It is clear that accounting for uncertainty in sighting observations is important as otherwise estimates of extinction time will be biased (Roberts et al. 2010). However, it is important to also recognise potential limitations of what can be determined from the sighting record. In particular, Solow and Beet (2014) introduce two plausible statistical models for a sighting record, the first assumes certain and valid uncertain sightings arise from the same process, whilst the second assumes they arise from independent processes. Conclusions regarding the extinction of the Ivory-billed woodpecker were different depending on which underlying process was assumed. Deciding which model is more appropriate will depend on the natural history of the sighting record, i.e. how the sighting observations have been recorded. It is also worth noting that the conclusions are also sensitive to the choice of prior distributions for the rate parameter, $\Lambda$.

A number of papers have proposed incorporating measures of sighting reliability, for example, assessment of certain and uncertain sightings was further expanded in Thompson et al. (2013) which incorporated expert estimates for observing the species. This approach was further extended in Lee et al. (2014) to incorporate uncertainty in the priors and inclusion of sampling effort and observation reliability was explored in Jarić and Roberts (2014). Numerical implementation of this approach can be found in Lee (2014). Thompson et al. (2017) provides guidance for simultaneous analysis of both record and survey data whilst accounting for data quality. Thompson et al. (2019) implements Bayesian updating, which allows probabilities of extinction to be updated year-by-year as new data come to hand. This approach means that the posterior distribution from year $t$ is set to be the prior in year $t + 1$ and results in a Bayes Factor which also updates each year. However, Lee et al. (2017) evaluated how useful expert opinion on the quality of sighting data is when inferring extinction and found that incorporating expert estimates of sighting reliability had little effect on inference, and instead classifying sightings as certain or uncertain was more useful. It should be noted that validity probabilities attached to sightings combine prior information about extinction time and information about the skill of the observer and so should not be used in isolation (pers. comm. Solow) when inferring extinction time (Lee et al. 2015).

An alternative approach proposed by Brook et al. (2019) obtains the empirical sampling distribution of extinction time, by re-sampling without replacement from the sighting record. Sighting reliability is used as the probability of inclusion of an individual sighting observation and extinction time is estimated using one of the preferred models described in this review. The confidence bounds of extinction time can be obtained directly from the empirical distribution. Solow (2023) documents some concerns regarding the statistical properties of the extinction date estimator and confidence interval proposed in Brook et al. (2019) and highlights that the estimated extinction time will exceed the true extinction time when reliability ratings are high. However, as described in Jarić et al. (2023) the method of Brook et al. (2019) is not designed to be a method to obtain an extinction-date estimator, rather it is proposed as a pre-processing step applied to the input data prior to further analysis. The re-sampling method does not make any assumptions about the model used for inference; thus, it can be used in combination with the methods presented in Table 1.

### Important model considerations

This article reviews the plethora of methodological developments which have been proposed to be able to infer species extinction from sighting data. With such a range of methods available, it is of course desirable to provide guidance on which methods should be used for a particular application. Each of the described methods will make model assumptions and it is unfortunately impossible to test the validity of these assumptions from sighting data alone. However, given conclusions from inference might vary depending on the method used, within this section we highlight aspects that should be considered before selecting a model. Typically, we would not endorse using multiple methods for a single case study – selection of the method should be determined by an understanding of how the sighting data has been recorded and the translation of that into the best mathematical interpretation of that process. For example, whether the sighting rate can be considered to be proportional to population size, which will be declining if a species is approaching extinction, or whether it is necessary to account for

uncertainty in observations. Within this section, we discuss aspects which should be considered when establishing which methodology to consider.

### How many sightings are required?

The OLE model of Solow (2005) requires the choice of $k$. The method necessitates that the joint distribution of the $k$ most recent sightings is the Weibull extreme value distribution and if $k$ is too large the asymptotic argument leading to the Weibull model may not hold. In contrast, if $k$ is too small issues of small sample size will arise and the inferential method will lack power. In his discussion Solow (2005) suggested that $k$ should be between 5 and 10. However, Clements et al. (2013) found, based on simulations from microcosm experiments, that accuracy increased as the size of $k$ increased, however, considerations of the validity of the Weibull model have not been explored, and thus conclusions are limited to the scenarios considered within the article. Most published work continues to follow Solow (2005) in the use of the most recent 5 to 10 sightings. It is, important to note that this does not apply to other models. For example, the non-parametric model of Solow and Roberts (2003) is based on the last two sightings, while other models such as Solow (1993a) will still generate a result when $n = 1$. Roberts and Jarić (2020) looked at the issue of inferring extinction for species known from only a single specimen, this is expanded on in the discussion.

### Sighting independence

While these models use sightings of a species, consideration needs to be given to the issue of non-independence in sightings. If there are several sightings of the species on the same date from the same location, then these could still be considered as one sighting. Whereas if the date and/or location were different then these could be considered separate sightings. In order to account for frequency data, rather than binary sighting observations, Burgman et al. (1995) demonstrated how the sighting period (0, $T$) can be partitioned into equal-sized units of time, with data now corresponding to 0, 1 or multiple sightings within each unit. Calculation of probabilities corresponding to the longest run of units with no sightings, conditional on the observed data, can then be used to infer extinction. The gain in statistical power of using frequency data rather than binary data is examined in Burgman et al. (2000).

A special case of extinction is the eradication of invasive species, where interest lies in inferring when a species has been eradicated. An important difference here is that often sightings used in these cases are extermination events (e.g. trappings) and therefore these sightings are non-independent of the decline of the species. While a number of models have been developed and applied in relation to the question of eradication (e.g. Regan et al. 2006; Solow et al. 2008; Rout et al. 2009; Ramsey et al. 2023), this issue needs to be borne in mind when also looking at certain extinctions such as the Thylacine (*Thylacinus cynocephalus*) where, in this case, bounty records may form a significant proportion of the sighting record.

### Multiple sightings within a year

Assuming all sightings are independent and would be used in the analysis, one issue that may arise is when there are multiple sightings in a single year. In some models, such as Solow (1993a), where the parameters used are $t_n$, $T$ and $n$, this is not an issue. However, in other models (e.g. OLE), the time between each sighting forms part of the model and these times cannot be zero. Where a more precise date is known then a decimal figure for a year can be used (e.g. for the date August 2022, the date used in the

model would be 2022.67). In the case when only the year is present, but the locations are different and therefore can be considered independent sightings, then a simple solution is to split the distribution of the sightings evenly across the year, such as 2022.0 and 2022.5.

### Choosing a start date

Choosing a start date needs to be carefully considered. For some, it may be obvious, such as the starting date of a long-term survey, or more arbitrary, such as a convenient date (e.g. 2000). In some cases, the first sighting, $t_1$, of the sighting record may be used. In this case, the starting point of the sighting record is non-independent of the sightings. Statistically, you are conditioning on the first sighting observation, and therefore, the number of sightings reduces by 1 to become $n - 1$.

Interestingly, for the uniform and truncated exponential models, it is statistically valid to set the time of the first sighting to be the start date since the subsequent sighting rates continue to be uniform or exponential. Furthermore, the truncated exponential estimator is based on a technical result that requires the minimum value to be 0, hence suggesting setting the initial observation to be at time 0 (Solow 1993b). Both methods are invariant to changes in scale, i.e. changes in the unit of measurement of time. However, if other parametric models for sighting rate were derived it may be necessary to incorporate an additional parameter within the model to account for the unknown start time (Solow 2005).

### Separation of population abundance and sampling effort

Caley and Barry (2014) propose a hierarchical Bayesian framework to separate the two processes of underlying population dynamics and the sighting process. They achieve this through the estimation of survival and detection probabilities conditional on the observed sighting data. The advantage of this approach is that it is not constrained to consideration of constant or declining population densities and indeed they have demonstrated how survival and detection probabilities can be density dependent. Through application to red fox carcass data, they have shown that there is more uncertainty on population persistence when the population process is made more complex and have used this to urge caution over conclusions drawn from the naive use of simplifying assumptions. A similar approach was considered in Kodikara et al. (2020) who in addition, extended the formulation to account for certain and uncertain sightings.

## Discussion

Although the methods reviewed here are based solely on sightings that are often seen as the minimum data that exists for a species, it is important to note that for many species even this level of data is scarce, with some species being only known from a single museum specimen. For example, Roberts and Jarić (2020) looked at the issue of inferring extinction in Malagasy orchids that are data-poor, where 31% of the species are known from a single herbarium specimen. As Roberts et al. (2016) showed in their study of data accumulation in Malagasy orchids, those species that are known from very little data, are not as such data deficient, but rather the lack of data is an indication of their rarity.

In general, sighting data contains temporal and spatial information, with temporal data having been made use of in the methods discussed here. The associated spatial element of the data has been rarely used other than when focusing on a geographically defined

area. More recently, Brook et al. (2023) reconstructed and mapped the spatio-temporal extirpation and eventual extinction of the Thylacine (*T. cynocephalus*). There is much work to do regarding the incorporation of the spatial element in modelling extinction. While this may go beyond the question of inference of species extinction, it is relevant in relation to local extirpation events and may provide useful evidence regarding the current known range of the species, even if it is not extinct.

The majority of models presented in this article assume that the sightings can be modelled as realisations of a point process and are thus continuous-time models. New methodological approaches which have already been discussed that are within a discrete-time framework include Thompson et al. (2013), Lee (2014), Lee et al. (2014). Alroy (2014) derived a discrete-time model, evaluating sequential posterior probabilities of extinction conditional on observations up to a given time. This differs from the methods presented in Inferring extinction section of this article as inference is based upon the entire sighting record (Solow 2016c). Consideration of whether to use a discrete or continuous time model should account for an understanding of the types of sightings that are recorded within the available sighting record. All sighting records will be observations recorded at discrete times; however, it can be appropriate to model these observations using a continuous time model, particularly in a situation when the observation period is long, and the time intervals between observations are short.

Beyond biological extinctions, these methods have now been applied to other extinction contexts, such as the end of the Acheulean stone tool culture (Key et al. 2021). Likewise, as these methods are interested in endpoints, they have the potential to be applied to infer the start, or origination, of processes, such as dating the origin of Covid-19 (Roberts et al. 2021) or the origins of the first flaked stone technologies (Key et al. 2021). It is, however, important to note that in these cases interest lies in dating a known extinction or origination events rather than answering the question as to whether a species is or is not extinct.

Much of the work in the extinction modelling literature reviewed here has largely focused on the theoretical development of inferential approaches, with associated model selection mainly based on how well model assumptions are perceived to fit the data. However, due to the type of data (i.e. often sparse sightings), knowledge of the extent to which the data fits the assumptions is poor. A number of papers (Rivadeneira et al. 2009; Clements et al. 2013) have undertaken power analyses to better understand the impact of factors such as varying sighting effort, population decline and violation of distributional assumptions underpinning on the performance of the methods. Vogel et al. (2009) used L-moment diagrams and probability plot correlation coefficient hypothesis tests to evaluate the goodness-of-fit of a number of models. However, it is still the case that it can be difficult to justify a particular approach for any specific application. Because of this, it is likely that new methods will emerge as a consequence of motivating properties of sighting data, for example, community-based data in marine settings as presented in Smith and Solow (2011). To select an appropriate method, it's essential to understand the underlying generation process of the sighting record, as outlined in the Important model considerations section. If the model's assumptions about how the data is generated do not match the sighting record, it could lead to inaccurate estimates of extinction.

An interesting extension of the question as to whether a species is extinct, is the question of functional extinction. The term functional extinction has often been confused, however Jarić (2015) recently provided a deeper understanding of this concept and the different forms it takes. In particular Jarić (2015) explained that functional extinction could be described as: (i) a population decline that leads to a loss of ecosystem services that the species provided or to a negligible contribution of a species to ecosystem processes; (ii) populations at very low abundances; or (iii) populations which experience time-delayed deterministic extinction which occurs due to persistent lack of reproductive success or recruitment. Each of these definitions will require different assessment to establish whether functional extinction has occurred. This has led to a number of papers for the inference of functional extinction (Jarić et al. 2016; Roberts et al. 2017; Zhang et al. 2020). These methods again use sighting data, although this may take the form of aged individuals that are sighted to give a timing of last breeding (e.g. Zhang et al. 2020), or a comparison of the sighting record of the species and the sighting record of the functional event such as nesting (e.g. Roberts et al. 2017). The modelling approach uses a population dynamic model to detect functional extinction on a sighting record of individuals of known or estimated ages. A detailed review of these methods is however beyond the scope of this article.

While this article focuses specifically on the use of sighting data in temporal models, often there is additional information that can be used in the inference of extinction. As mentioned, the criteria for the category of Extinct, as laid out by the IUCN Red List (IUCN 2012), emphasises other aspects such as that the surveys that have been carried out are appropriate to the species' biology (i.e. correct time, season, habitat, etc.). Thompson et al. (2019) have gone some way towards incorporating these elements into a model for inferring extinction. In their study into the attributes of extinction declarations, Roberts et al. (2023) surveyed expert assessors from the IUCN Species Survival Commission's Specialist Groups using a choice experiment approach. They found that in the main, data availability, time from the last sighting, detectability, habitat availability, and population decline were all important attributes used when inferring extinction, although there were slight differences between certain groups. While Keith et al. (2017) provided a framework for the inference of extinction based on a qualitative approach using expert judgement. This qualitative approach incorporated mapping and structured elicitation.

As a result, the temporal models discussed here should only be considered as one line of evidence when making extinction declarations. Future extensions of these models should consider how other attributes of extinction can be incorporated to aid with the robust assessment of extinction. A first starting point may be to consider how we can infer attributes such as detectability, habitat availability, etc., but that we must consider for the majority of species data is scarce.

**Open peer review.** To view the open peer review materials for this article, please visit http://doi.org/10.1017/ext.2024.18.

**Data availability statement.** The methods presented in Table 1 have been collated in an RShiny app, available at https://tommy-cheale-shinyapps.shinyapps.io/New_ext_pap/.

**Acknowledgements.** We thank the Editor, Handling Editor, Andrew Solow and one anonymous reviewer for their detailed comments on our article. Their recommendations have greatly improved the article.

**Author contribution.** RSM reviewed the statistical methodology, TC wrote the RShiny app, ECF explored the mathematical underpinnings of the approaches and DLR reviewed the contextual and main developments of this field. All authors contributed to the writing of the article.

**Financial support.** RSM and DLR were funded by the British Academy, Royal Academy of Engineering and Royal Society Academies Partnership in Supporting Excellence in Cross-disciplinary research award (APEX award, AA21/100175). RSM was supported by Leverhulme Research Fellowship RF-2022-197. TC is a PhD student funded by EPSRC grant EP/W524050/1.

**Competing interest.** DLR is a senior editor of the journal *Cambridge Prisms: Extinction*.

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
