## [Editor Report]

Firstly, apologies for the lengthy time that your manuscript was in review. But the two reviews we now have in are positive and have a modest set of suggestions about how the ms can be revised and improved - these will, nonetheless, need careful thought and serious effort to implement, as they are all important points. In particular, R1 suggests a fundamental reworking of the focus of the review to emphasize more recent developments since 2015, when another major review on this topic was published. My feeling is that every review article should stand on its own and summarize the field up to that point in time, so you still should offer an overview of earlier literature to provide a good historical background to the development of the topic. But I agree with R1 that a greater emphasis on more recent work that is as yet unreviewed is desirable - I’m sure you can find the right balance to strike. Another of the reviewer’s points I’d like to emphasize is R2’s suggestion to explain mathematical models in words as well as in maths - this is important to maximize accessibility of your message to those who are less mathematically literate than others.

---

## [Editor Report]

One of the reviewers suggests a few more small additions - these should be easy to include and will put the final touches on your paper.